# Constructing a Hierarchical Hydrophilic Crosslink Network on the Surface of a Polyvinylidene Fluoride Membrane for Efficient Oil/Water Emulsion Separation

**DOI:** 10.3390/membranes13030255

**Published:** 2023-02-21

**Authors:** Ruixian Zhang, Yuanbin Mo, Yanfei Gao, Zeguang Zhou, Xueyi Hou, Xiuxiu Ren, Junzhong Wang, Xiaokun Chu, Yanyue Lu

**Affiliations:** 1Guangxi Key Laboratory for Polysaccharide Materials and Modification, Guangxi Higher Education Institutes Key Laboratory for New Chemical and Biological Transformation Process Technology, School of Chemistry and Chemical Engineering, Guangxi Minzu University, Nanning 530006, China; 2Institute of Artificial Intelligence, Guangxi Minzu University, Nanning 530006, China

**Keywords:** superior separation method, hierarchical superhydrophilic network, steady polyphenol coating, oil-in-water emulsion

## Abstract

Oil/water mixtures from industrial and domestic wastewater adversely affect the environment and human beings. In this context, the development of a facile and improved separation method is crucial. Herein, dopamine was used as a bioadhesive to bind tea polyphenol (TP) onto the surface of a polyvinylidene fluoride (PVDF) membrane to form the first hydrophilic polymer network. Sodium periodate (NaIO_4_) is considered an oxidising agent for triggering self-polymerisation and can be used to introduce hydrophilic groups via surface manipulation to form the second hydrophilic network. In contrast to the individual polydopamine (PDA) and TP/NaIO_4_ composite coatings for a hydrophobic PVDF microfiltration membrane, a combination of PDA, TP, and NaIO_4_ has achieved the most facile treatment process for transforming the hydrophobic membrane into the hydrophilic state. The hierarchical superhydrophilic network structure with a simultaneous underwater superoleophobic membrane exhibited excellent performance in separating various oil-in-water emulsions, with a high water flux (1530 L.m^−2^ h^−1^.bar) and improved rejection (98%). The water contact angle of the modified membrane was 0° in 1 s. Moreover, the steady polyphenol coating was applied onto the surface, which endowed the membrane with an adequate antifouling and recovery capability and a robust durability against immersion in an acid, alkali, or salt solution. This facile scale-up method depends on in situ plant-inspired chemistry and has remarkable potential for practical applications.

## 1. Introduction

With the development of science and technology, environmental protection has been attracting extensive interest. Oil and water mixtures generated from oil spills and long-term industrial production significantly affect the environmental balance, endangering marine organisms and economic development [1,2,3,4,5]. Therefore, repairing the vulnerable ecosystem is critical [6,7]. Oil–water separation can be crucial in addressing the issues related to industrial oily wastewater and other types of oil–water pollution. Among the excellent technologies for separation, membranes with super wettability have attracted considerable research attention, owing to their low energy consumption and high efficiency [8,9,10,11,12]. Accordingly, the development of unique techniques that can separate various oil-polluted water streams with excellent recyclability and mechanical stability is paramount [7,12,13,14].

Recently, among the different traditional methods for separating surfactant-stabilised oil–water emulsions, membrane processes have been acknowledged as the most promising alternative separation medium because of their rapid separation, low energy consumption, and facile operating conditions, especially compared with air flotation [15,16], adsorption [17,18], coalescence [19], gravity settling, and coagulation [20,21,22]. Enhancing the interface hydrophilicity of membranes has shown excellent performance when dealing with oily wastewater [23,24]. Conventional strategies of surface modification, such as spin-coating [25,26], grafting [27], bio-adhesion [28], atomic layer deposition [29], and surface coating, have been considered to exhibit considerable efficacy [30,31]. Generally, the best renowned representative is considered to be surface coating via synthesis multifunctional wetting coating [32,33]. By exploiting surface coating, superhydrophilic membranes can circumvent a series of time-consuming processes and long-term operation [34,35].

In the past few years, extensive focus has been devoted to the landmark study of mussel-inspired hydrophilic modification materials [20,36,37]. Plant polyphenol-based coatings have garnered extensive research interest because of their stable surface binding affinity and drastic hydrophilicity on diverse substrates [38]. For example, Qiu et al. reported a straightforward co-deposition strategy based on a reaction between catechol and polyethyleneimine to form a constructed hierarchical superhydrophilicity layer on the surface of polypropylene microfiltration membranes. Their objective was to realise a time-effective decolourisation of anionic dye sewage [39]. Zeng et al. developed a plant polyphenol-inspired PA@PEI polyelectrolyte complex, integrating the high negative charges of phytic acid (PA) and the powerful cationic charge density of polyethyleneimine (PEI) in aqueous solution via a one-step self-assembly approach with excellent static and dynamic water stability [40]. Moreover, based on the catechol reaction, Wang et al. proposed a common approach for depositing tannic acid (TA) with a distinct adhesion property and the hydrolysis product of 3-aminopropyltriethoxysilane (APTES) in aqueous solution to form hydrophilic and hierarchical layered colloidal nanospheres [41]. Therefore, plant polyphenol superhydrophilic coatings are expected to significantly expand the application scope of water remediation and recycle.

Based on this background, we fabricated a hydrophilic PDA/TP/NaIO_4_ coating on the polyvinylidene fluoride (PVDF) membrane surface. Consequently, utilising the robust coating of polydopamine (PDA), we produced a superhydrophilic and hierarchical surface structure on the porous substrate via covalent and noncovalent interactions to reduce the membrane pore size and increase the separation effect. Additionally, we exploited tea polyphenols (TPs) and sodium periodate (NaIO_4_) to optimally assemble a uniform hydrolysable polyphenol skin layer by polyphenol-coating reaction kinetics. The economical composite membrane exhibited not only sufficient resistance toward different types of oil/water emulsions under an ultralow vacuum filtration condition (0.01 MPa) but also considerable chemical stability regardless of the solution PH.

## 2. Experimental

### 2.1. Materials

Commercial PVDF microporous membranes (mean pore size = 0.22 um) were procured from Shanghai Mosu Science Equipment Co. (Shanghai, China). TPs (98%) were procured from Shanghai Yuanye Bio-Technology Co., Ltd. (Shanghai, China). Sodium periodate (NaIO_4_, 98%); ethanol (EtOH, AR); tris (hydroxymethyl)-aminomethane (Tris-HCl, 1 M, pH = 8.5); dopamine hydrochloride (DA, ≥98%); sodium dodecyl sulfonate (SDS); petroleum ether; n-hexane; n-heptane; toluene; and trichloromethane were procured from Shanghai Aladdin Reagent Company (Shanghai, China). Deionised (DI) water was used in all experiments. All chemicals were used as received.

### 2.2. Preparation of Biocomposite Membranes

First, the unmodified PVDF membranes were successively pre-wetted by ethanol and deionised water to remove surface foulants and distract the pores before soaking in 50 mL of the PDA Tris-HCl buffer (1 M, pH = 8.5). Subsequently, the as-prepared membranes were transferred into TP (20 mg/mL in the buffer solution) for 5 min and dissolved in 50 mL NaIO_4_ (20 mg/mL in the aqueous solution) for another 5 min. Thereafter, the samples were cleaned by DI water overnight to scrub off unstable residues and dried in vacuum at 35 °C. For comparison, the nascent PVDF membranes were dipped into a solution containing only PDA for the same time as that in the aforementioned step. Moreover, the composite membranes coated only by TP and NaIO_4_ obtained using the same method above were used to compare the functionalisation of the PVDF membrane.

The abbreviation M0 refers to pristine PVDF microporous membranes. The membranes prepared only by PDA or TP/NaIO_4_ were denoted M1 and M2, respectively. Finally, the PDA/TP/NaIO_4_-coated membranes were labelled M3.

### 2.3. Characterisation

The surface morphology of the composite membranes was observed by scanning electron microscopy (SEM, JIEKE TESCAN MIRA LMS, Shanghai, China) with gold spraying pre-treatment. X-ray photoelectron spectroscopy (XPS, ThermoFischer, ESCALAB 250Xi, Waltham, MA, USA) and attenuated total reflectance Fourier transform infrared spectroscopy (ATR-FTIR, Nicolet iS 10, Waltham, MA, USA) were conducted to analyse the chemical composition and surface functional groups. A three-dimensional atomic force microscope (AFM, Bruker Dimension ICON, Germany) was used to determine the surface roughness. Water contact angles (WCAs) and underwater oil contact angles were recorded by a contact-angle-measuring instrument (OCA, SDC-350, Guangdong, China). The filtration experiments were validated by vacuum filtration equipment. The magnified views of the feed and permeate solutions of the oil-in-water emulsion were observed by an optical microscope (ML31-P MS60, Guangdong, China). An ultraviolet spectrophotometer (UV-3600, Hangzhou, China) was used to analyse the oil content of the emulsion.

### 2.4. Separation Test of Oil/Water Emulsions

Briefly, 1 mL of oil (petroleum ether, n-hexane, n-heptane, toluene, and trichloromethane) was mixed with 99 mL water; thereafter, 0.01 g of SDS was added as an emulsifying agent and stirred at 1000 rpm for >6 h to obtain a surfactant stabilised oil-in-water emulsion.

The prepared membranes were sealed by the laboratory vacuum filtration pump at a stable pressure (0.01 MPa) and room temperature. Subsequently, the modified PVDF membrane was pre-compacted by pure water until a stable value was reached. The emulsion permeability was assessed by recording the real-time flux (*F*), and its rejection (*R*) was measured using a UV spectrometer. The flux and rejection were defined as:(1)$$F=\frac{V}{A\times t\times P}$$
(2)$$R=\left(1-\frac{C_{p}}{C_{f}}\right)\times 100\%$$
where *V* (L) is the volume of the permeated liquid; *A* (m^2^) is the area of separation; *t* (h) is the filtration duration; *P* (bar) is the transmembrane pressure; *C_p_* and *C_f_* denote the oil concentrations in the filtrate and feed solutions, respectively.

### 2.5. Stability and Anti-Oil Fouling Performance

Considering practical applications, we calculated the variations in permeate flux and rejection under an ultralow applied transmembrane pressure of 0.01 MPa by a long-term cycle emulsion separation test. Before the next treatment, the membrane was simply rinsed with deionised water. Furthermore, the membranes were immersed in aqueous media of various pH levels and saline (saturated sodium chloride) to evaluate the chemical and physical stability of the hydrophilic coating through the contact angle of the sample.

## 3. Results and Discussion

### 3.1. Surface Chemistry of the Superhydrophilic Membranes

ATR-FTIR and XPS were conducted to investigate the surface chemical components and elemental distribution of the as-constructed PVDF membranes. In Figure 1a, the strong infrared absorption at 1398.7 cm^−1^, 1100.3 cm^−1^, and 890.2 cm^−1^ is attributed to the PVDF membrane. Unlike in the case of the pristine membrane, the new stretching vibration in the range of 3100–3600 cm^−1^ and the marginal absorption peak of aromatic C=C resonance vibrations at 1610 cm^−1^ are assigned to the phenolic hydroxyl groups owing to the successfully deposited PDA. The self-assembly of DA could increase the polar functional groups and hydrophilic, which can be confirmed by the variation in the contact angle (to be described later). Nevertheless, the additional peaks at 1740 cm^−1^ are ascribed to the carboxyl absorption peak, which is also the case for membranes treated by the oxidising reaction of TP and NaIO_4_.

As shown in Figure 1b, the results of the XPS survey scan profile are consistent with those of the FTIR profile. For the substrate membrane, the prominent peaks at 289.89 eV and 701.25 eV are assigned to C1s and F1s, respectively. With the surface modification approach, two apparent peaks of O1s and N1s observed at 545.25 eV and 403.89 eV, respectively, accompanied by the decreased intensity of the F peak, demonstrate the successful construction of the hydrophilic polymer network.

The PVDF membrane has been considered a base skeleton by virtue of its excellent mechanical strength and chemical tolerance [42]. In a weakly alkaline environment, dopamine could deposit a polydopamine layer on the PVDF surface via self-polymerisation, providing robust adhesion stability (Figure 1a). Importantly, the formed PDA membranes can further provide reactive sites to accelerate secondary reactions [43]. After immersion in the alkaline incubation solution, TP was slightly oxidised to yield o-quinine, which could covalently attach with the amino and catechol groups from the PDA molecule by mutual Michael addition or the Schiff base process [44]. After this step, TP was evenly and strongly grafted onto the membranes due to the powerful linking of PDA with the porous surface via covalent and noncovalent bonding. Subsequently, NaIO_4_ served as an oxidising agent to further expedite the deposition of TP and production of carboxyl function groups, which markedly reinforced the hydrophilicity of the modified membranes (Figure 1b). The making process of the composite membrane was described in Figure 2.

Furthermore, according to Table 1, compared with other membrane samples in some literatures, the PDA/TP/NaIO_4_ membrane exhibited an incredibly short period of modified time and extremely low operating pressure, which invested as-developed membrane with a promising candidate for the separation of oil–water mixtures in actual application.

### 3.2. Surface Morphology

The magnified SEM images reveal the morphology and differentiation of the surface and cross section, respectively. As shown in Figure 2a, the pristine membrane exhibits a relatively smooth skeleton and a clean, porous surface structure. Figure 2b shows that upon PDA coating, a rough construction is observed, where a mound of nanoparticles is conformally aggregated on its surface and internal pore channels, leading to narrow pore size distributions. Further, the PDA coating not only enhances the bonding force between the coating layer and substrate but also increases the surface wettability derived from hydrophilic groups. Figure 2c illustrates a distinct interconnected micropore skeleton, which can be ascribed to the oxidation reaction between TP and NaIO_4_ producing a rough hydrophilic coating on the substrate, further improving the film wettability properties. The SEM images in Figure 2d show the excessive amounts of PDA nanometre bumps stuck on the surface of the PVDF membranes as the mesosphere; furthermore, they show cross layers connected with each other to form a biomimetic hybrid net, which optimise permeability.

The cross section and roughness of the pristine and modified PVDF membranes are displayed in Figure 3. Clearly, the base and modified membranes have a sponge-like microstructure. The thickness and roughness of the pristine PVDF membrane are 114 ± 0.5 µm and 152 µm, respectively (Figure 3a). With the clustered PDA particles, the membranes exhibit the tendency to thicken (Figure 3b,d). The Wenzel and Cassie models indicated that the increase in the surface roughness of the micro/nanostructure is accompanied by the amplification of the intrinsic hydrophilicity of the surface [14,48]. Therefore, combining the coating of PDA and TP/NaIO_4_, the composite membrane augments the roughness of surfaces (Ra = 335 µm) and forms micro-nano structures, allowing water molecules to quickly pass through. This is in accordance with the subsequent variation in water flux.

### 3.3. Wetting Behaviours and Water Flux

The benign wetting behaviours and water flux of the membrane are the vital basis for estimating whether water can transport and enter the filtration channel quickly. Utilising the WCA and underwater oil contact angle (UOCA), we can assess the wetting performance of the membranes. Figure 4a indicates that the WCA and UOCA of the pristine membrane are 117° and 66°, respectively, owing to its hydrophobicity. So, the rejection rate of chloroform emulsions is the lowest (Figure 4b). The corresponding pure water flux reached 1675 L/m^2^·h·bar. Moreover, clear droplets can be observed on the membrane surface (Figure 5a). Upon surface coating, the M1 and M2 samples gradually transformed to a hydrophilic state, whereas the decrease in pure water flux was ascribed to the loading of the surface hydrophilic coating net and aggregate particles. The initial contact angles of M1 and M2 are 99° and 45°, respectively, and the hemispherical droplets are all visible on the surface (Figure 5a). The UOCAs of a range of membranes and oils were measured by a contact angle goniometer to detect a superior underwater repellence. In comparison, the extreme value variation of M3 from WCA to UOCA is >150°, demonstrating the excellent wetting and antifouling behaviour (Figure 4a,b). Therefore, the rejection rate of chloroform emulsions is higher than M1 and M2 membranes (Figure 4b). Figure 5a indicates that when water drops on M3, it is immediately wetted with a WCA of 0°.

By the combination of PDA/TP/NaIO_4_, M3 exhibited adequate superoleophobicity (UWCA > 140°) toward all selected oils, including petroleum ether, n-hexane, n-heptane, toluene, and trichloromethane (Figure 4c). The larger the UWCA, the better is the anti-adhesion work on the solid surface. The adhesion resistance of the membrane was verified by chloroform (Figure 5b), where the oil drop did not stick to the film, irrespective of the pressing force. To summarise, M3 was successfully endowed a composite coating layer by co-deposition.

### 3.4. Resistance to the Oil/Water Emulsion

The superhydrophilicity and underwater superolephobicity could be attributed to the mutual effect of the polymeric network and nano hierarchical surface structure, which enhances the water affinity and oil fouling resistance. Five types of surfactant-stabilised oil-in-water emulsions, namely, methyl benzene-in-water, chloroform-in-water, petroleum-in-water, n-hexane-in-water, and n-heptane-in-water, were operated at a transmembrane pressure of 0.01 MPa to evaluate the water flux and oil rejection of the PDA/TP/NaIO_4_ membrane. The oil content in the filtrate was detected using a UV spectrophotometer. Before the experiment, M3 was pre-moistened with pure water, following which the measured amounts of the various oil–water emulsions were poured into the vertical separation device, which was driven by a vacuum pump to allow the water to pass through. Because of the dynamic hydrophilic surface coating, M3 displayed excellent separation flux values ranging from 891–1530 L/m^2^·h·bar. As shown in Figure 6a, the petroleum emulsion exhibits the highest permeation flux, which can be attributed to its low density and viscosity. Furthermore, the as-prepared membrane exhibits an ultrahigh oil retention rate of 95.2–97.2% throughout the experiments.

Generally, the underwater oleophobicity is proven by the oil drop injection test. As demonstrated in Figure 6b, the raw PVDF membrane surface is contaminated by a jet of stained chloroform solution. By contrast, the modified membrane has a repulsive effect on oil, and the oil leaves no trace on the surface, which is attributed to the construction of multi-scale rough structures. Hence, it is beneficial to capture water on the surface and reduce the contact area between the membrane and oil.

The optical micrographs of five diverse types of oil–water emulsion feed and permeate are shown in Figure 7. Before separation, all emulsions were dyed red for easy identification. The photographs show that numerous oil drops dispersed unevenly. After filtration, the filtrate underwent clarification and most oil droplets were successfully removed, demonstrating comparatively high separation efficiency. Notably, the different emulsions had droplet size distributions ranging from 2.5 to 4.5 µm, which was larger than the average pore size of the membrane.

### 3.5. Mechanical Durability

To investigate the peculiarity of the hydrophilic layer in various harsh environments, including acid, aqueous, and salty solutions, as well as long-time sonication, the stability and recyclability performance of as-prepared membranes were evaluated. Figure 8a–e shows that the permeated flux of multiple oil-in-water emulsions including methyl benzene, chloroform, petroleum ether, n-hexane, and n-heptane show a slight decline after five rounds of cyclic experiments. The membranes were simply cleaned with water after each individual filtration cycling test to wash away suspended oil droplets on the surface, and all types of oil-in-water emulsions were fed through the membranes under a steady pressure of 0.1 bar. The decreased permeability values ascribed to the foulant were retained and accumulated, forming an oil cake layer and blocking parts of the pores. Nevertheless, oil repellent behaviours were estimated by rejection and no distinct decline was shown throughout filtration. The average oil rejection fundamentally remained at 94–96%, regardless of the oil variant, confirming that the polyphenol composite membrane exhibits satisfactory separating and robust antifouling properties.

Generally, the hydrophilic surface is easily damaged in hostile conditions, such as extreme-pH aqueous media, a NaCl solution, and sonication treatment [49]. The variation in contact angle is a key indicator of the mechanical stability of the PDA/TP/NaIO_4_ coating. The representative curves of M3 CA declining with time are shown in Figure 8f. After 6 h of incubation, the as-prepared membrane maintains its original wetting peculiarity with no noticeable decrease in the WCA (CA = 0), even though the surface colour was mildly bleached in the erosive media. Polydopamine, usually considered to be a dual-function platform spacer, was speculated to reinforce the nanomaterial adhesion force and stability of the coating [50]. The noncovalent interaction between PDA and the commercial membrane provides durability and reusability.

## 4. Conclusions

A facile, mussel-inspired, high-performance PVDF superhydrophilic membrane was fabricated for the effective separation of various oil-in-water emulsions. The salient modified strategy was applied by dip-coating PDA/TP/NaIO_4_, which deposited a multilayer mussel-functionalised network onto the surface of the target substrate. The as-prepared membrane exhibited appreciable water permeability under ultralow transmembrane pressure. The steady coating anchored on the surface was imparted with remarkable repulsion of oil and significant mechanical stability during the cycle test. With these features, the modified hydrophilic membrane displays considerable potential for application in oil–water treatment.

## Data Availability

The data used to support the findings of this study are available from the corresponding author upon request.

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
