# Peer review of "Constructing a Hierarchical Hydrophilic Crosslink Network on the Surface of a Polyvinylidene Fluoride Membrane for Efficient Oil/Water Emulsion Separation"

_membranes, 2023, doi:10.3390/membranes13030255_

Round 1

Reviewer 1 Report

The authors use dopamine as an adhesive to bind TP onto PVDF to form a hydrophilic membrane for oil/water separation. The work is interesting and well-presented. I would recommend publication after minor revisions.

My only comment is that it seems that the durability of the hydrophilic membrane is not so good. Separation efficiency starts to drop starting from the second cycle, depending on various oils. Are there SEM images available for membranes after a few cycles? What could be ways to improve the membrane durability?

Author Response

Dear editor

On behalf of my co-authors, I would like to thank you very much for giving us the opportunity to revise our manuscript and I am very grateful to the editor and reviewers for their positive and constructive comments and suggestions on our manuscript entitled "Constructing a hierarchical hydrophilic crosslink network on the surface of polyvinylidene fluoride membrane for efficient oil/water emulsion separation".

First, response to comment: Are there SEM images available for membranes after a few cycles?

Response: We apologize for this oversight and will be submitting SEM images later.

Second, response to comment: What could be ways to improve the membrane durability?

Response: Firstly, the PVDF membrane is hydroxylated by alkali treatment in ammonia before being introduced into the PDA. Secondly, we will reinforce the surface hydrophilic coating by bonding between PVDF and PDA under ultrasonic conditions.

These are my answers to the questions asked by the editors and reviewers. I hope that the revised content will be approved and thank you again for your comments and suggestions.

Reviewer 2 Report

This manuscript described the fabrication of surface-treated PVDF membrane and their application in water-oil separation. The treated membranes exhibited excellent hydrophilicity and superoleophobicity. The reported full characterization of the control (PVDF) and PDA-treated membranes. However, the oil rejection was only reported for the M3 membrane. The authors must report the oil rejection results of M0, M1, and M2 membranes.

Comments:

1.       The abstract indicates the membranes exhibit excellent performance in water with high water flux and improved rejection (98%). However, the water flux was lower than the control PVDF, and the control PVDF's oil rejection was not reported to claim the improved rejection.

2.       Why did the authors report the oil rejection of the M3 sample only, regardless of having the lowest water flux? Other membranes that have simpler treatment (M2) could have better performance.

3.       The oil concentration of the emulsion (~10,000 ppm) is very high. Such high concentration is easily removed by mechanical means. Membrane separations are necessary for lower concentrations (~ 100 ppm). With 98% rejection, the residual oil concentration is still high (200 ppm). The authors should demonstrate the modified membranes have a high rejection of low-concentration oil (50 – 200 ppm).

4.       The oil rejection of the control PVDF must be reported. The droplet size of the used oil ranges between 1 to 7.5 mm.  therefore, the 0.22 mm PVDF membrane should have good rejection.

5.       In line 244, the authors stated, “By the combination of PDA/TP/NaIO4, M3 exhibited adequate superhydrophobicity”. Do you mean superoleophobicity?
